# The Potential Importance of Social Capital and Job Crafting for Work Engagement and Job Satisfaction among Health-Care Employees

**DOI:** 10.3390/ijerph17124272

**Published:** 2020-06-15

**Authors:** Göran Jutengren, Ellen Jaldestad, Lotta Dellve, Andrea Eriksson

**Affiliations:** 1Department of Work Life and Social Welfare, University of Borås, SE-501 90 Borås, Sweden; goran.jutengren@hb.se; 2Division of Ergonomics, KTH Royal Institute of Technology, SE-141 57 Huddinge, Sweden; andrea4@kth.se; 3Department of Sociology and Work Science, University of Gothenburg, SE-405 30 Gothenburg, Sweden; lotta.dellve@socav.gu.se

**Keywords:** social capital, job crafting, well-being, health care, mediating effect

## Abstract

(1) Background: Both employees and organizations benefit from a work environment characterized by work engagement and job satisfaction. This study examines the influence of work-group social capital on individuals’ work engagement, job satisfaction, and job crafting. In addition, the mediating effect of job crafting between social capital on the one side and job satisfaction and work engagement on the other side was analyzed. (2) Methods: This study used data from 250 health-care employees in Sweden who had completed a questionnaire at two time points (six to eight months apart). Analyses of separate cross-lagged panel designs were conducted using structural regression modeling with manifest variables. (3) Results: Social capital was predictive of both job satisfaction and work engagement over time. The results also indicated that higher degrees of social capital was predictive of more cognitive and relational, but not task-related job crafting over time. There was no clear evidence for a mediating effect of job crafting for social capital to work engagement or job satisfaction. (4) Conclusion: It would be beneficial for the health-care sector to consider setting up the organizations to promote social capital within work groups. Individual workers would gain in well-being and the organization is likely to gain in efficiency and lower turnover rates.

## 1. Introduction

Employees who like their jobs and are engaged in their work tasks are likely to experience higher degrees of psychological empowerment [1], better health [2,3], a better sense of psychological coherence [4], and less compassion fatigue [5]. From an organizational perspective, having satisfied and engaged employees is associated with lower rates of personnel turnover [6,7], higher degrees of organizational commitment [8], better patient care quality [9], and increased work effectiveness [5,7]. In the context of a health-care system that faces challenges, such as persistent demands for a more streamlined organization, maintaining a heavy workload, increasing the quality of care [10], and difficulties in attracting and retaining health-care personnel [11,12], there are strong reasons to discover how job satisfaction and work engagement can be best promoted.

Our general aim in this study is to attain a better understanding of the ways the concepts of work engagement and employee job satisfaction possibly can be promoted by employees’ self-imposed behaviors intended to make their job fit better with their interests and motivations (i.e., job crafting) and by a positive psychosocial work climate characterized by mutual trust (i.e., social capital). These concepts have been frequently examined during the last decade, but there are yet no consensus as to how they influence each other.

### 1.1. Work Engagement and Job Satisfaction

Although related, the concepts of job satisfaction and work engagement differ in some distinct ways. Work engagement has been defined as a positive and fulfilling work-related state of mind that is characterized by (a) a sense of being fully and happily concentrated on one’s work (i.e., absorption), (b) feeling significant, enthusiastic, inspired, and proud in relation to one’s job (i.e., dedication), and (c) having high levels of energy and mental resilience (which may be defined as robustness against mental exhaustion) in the work environment, as well as being willing to invest personal effort and persistence in one’s work (i.e., vigor) [13]. Compared to work engagement, job satisfaction is more of a passive form of employee well-being [14]. Bakker and Oerlemans [15] uses a two-dimensional model of work-related subjective well-being to further distinguish work engagement and job satisfaction from each other. In this model, work engagement is defined by both pleasure and activation, whereas job satisfaction is described as a pleasant state, but also a passive one with low levels of activation. Employees with high engagement are excited, enthusiastic, energized, happy, and pleased. Employees who are satisfied with their work situation are content, relaxed, and calm. In this study, we define job satisfaction as a positive work-related state of mind where the employee is content with the overall work environment and outcomes [16]. To further point out the difference between the two concepts explicitly, work engagement, on one hand, can be described as the proactive psychological involvement in the job to achieve collective or individual goals [17]. Job satisfaction, on the other hand, can be described as a retrospective cognitive and/or emotional subjective evaluation of the overall job situation [18].

### 1.2. Social Capital and Its Potential Impact on Work Engagement and Job Satisfaction

Social capital has been studied in a variety of contexts, such as families, residential neighborhoods, and communities, and it has been defined in multiple ways. Most definitions, though, emphasize the elements of networks and norms [19]. Social capital is commonly seen as a group characteristic rather than an individual trait [20]. In this paper, we follow Kouvonen et al.’s [19] approach by defining social capital as a positive, trustful, and innovative learning environment within work groups that focus on organizational social capital in terms of horizontal social relationships between colleagues. From this perspective, social capital then consists of two different components—cognitive and structural—which are commonly covered when the concept is operationalized. The cognitive component refers to the individual’s values and norms and their perceptions of support, reciprocity, solidarity, and trust, whereas the structural component refers to group-based activities, such as transparent decision-making processes, mutual accountability, practices of collective action, and joint responsibility. In addition, structural social capital also includes individuals’ interactions with social networks that give them access to various resources, such as advice and social support. Hence, for individuals, social capital can be beneficial by having access to group resources [21,22]. At an organizational level, social capital can be seen as a resource for the common good of the organization through collaboration, mutual trust, and justice [23].

Empirical studies from Germany and Sweden indicate that social capital is a positive resource in organizations that can be associated with both individual work engagement [24,25] and job satisfaction [25,26]. The association between social capital and work engagement has also been supported in Japanese samples [27]. Two methodological features these studies have in common are their quantitative approach and their relatively large samples of hospital personnel. By using the same methodological approach, comparisons can be drawn and extrapolated to other work settings. Meng, Clausen, and Borg [28] performed a large-scale study with personnel in the Danish dairy industry, including both blue collar workers and managers. Their results showed that there was a consistent positive association between the bonding (i.e., within teams), bridging (i.e., between teams), and linking (i.e., between teams and management) aspects of social capital on the one side and work engagement on the other side. The strength of the Meng et al.’s [28] study was its high reliability and low attrition, and their analyses controlled for gender and random effects at the team level.

Although the reviewed empirical studies support an association between social capital and work engagement and job satisfaction, all except one [25] used cross-sectional designs. Strömgren et al. [25] studied the predictive effect of social capital across a one-year period. However, using conventional statistical methods (e.g., logistic regression), their results may have been inflated by temporal variance. In addition to the limited empirical evidence for longitudinal effects of social capital, researchers have been critical of social capital because of its potentially negative aspects, commonly referred to as “the dark side of social capital” [29] (p. 943). For example, there can be a negative impact that a dysfunctional but strong work group can have on intergroup and intragroup relationships. In other words, members of a work group who display mutual trust, respect, and full acceptance of each other, might, for instance, have their minds set on a collective aim that contradicts the purpose of the larger organization.

In sum, we found no convincing evidence in the literature of any predictive effects of social capital over time on either work engagement or job satisfaction. To establish a potentially predictive effect of social capital, there is a need for a longitudinal study that is designed to control for temporal variance, and our study fills this gap in the literature.

### 1.3. The Role of Job Crafting

Job crafting can be defined as employees actively redesigning their jobs in a bottom-up process, where individuals change the boundaries of their jobs to fit their own skill sets with the aim of creating the best fit between their own individual desires, resources, and job demands [30,31,32]. Individuals may craft the cognitive, relational, and task-related boundaries for their work with the intention of redefining the purpose of their duties and their own work-related identity to achieve a more appealing job situation [32]. Changing the cognitive boundaries of the job involves employees altering their perceptions of the job as to think of it as, for instance, an important part of a larger whole. For employees to craft their relational boundaries, they must alter both whom to interact with and the nature of job-related interactions. Task-related crafting includes, for example, adding extra tasks to the job or changing types of tasks.

There are both conceptual and empirical reasons to believe that social capital predicts job crafting over time. As suggested by job crafting theory [32], motives for employees to craft their jobs are related to a perceived need to change some aspect of their work. The results of a recent metasynthesis of 24 qualitative studies of job crafting support this suggestion by indicating that employees did report attempts to improve their self-image, to connect with colleges, and to gain control over their work as motives for job crafting [33]. In addition, the most noteworthy motives identified in the same metasynthesis were closely related to a desire to perform well. Lazazzara and colleagues [33] also regard social support as part of a supportive context that may enhance forms of job crafting (e.g., adding extra tasks, and caring moves). From the perspective of Kouvonen and colleagues’ [19] definition of social capital, which emphasizes mutual trust among colleagues, one might argue that social capital (in line with social support) provides a milieu that allows individuals to try out and learn new and more expedient procedures that aligns to both their personal preferences and their intent regarding their work. As individual traits, such as a proactive personality, extraversion, and general self-efficacy, are positively correlated with job-crafting behavior [34], one might argue that higher degrees of social capital may facilitate initiatives of job crafting for the majority of employees.

Although there are few empirical quantitative studies on the association between social capital and job crafting, a study by Qi, Li, and Zhang [35] provides some support. They surveyed 220 employees in a manufacturing company in China using self-report instruments with good reliability and found a rather strong association between social capital and general job crafting. However, because of the cross-sectional design, the association that was reported does not provide evidence for a causal association in any direction.

Job crafting theory implies that employees’ intention to craft their job is related to them striving to improve their work experience [32,36]. There are also convincing empirical evidence that actual job-crafting behavior is directly associated with both work engagement and job satisfaction. In a recent meta-analysis, Rudolph et al. [34] examined associations between job crafting and various work outcomes. They found that job crafting intended to increase beneficial job characteristics, rather than decreasing hindering job demands, was moderately to highly correlated with work engagement and moderately correlated with job satisfaction. Lichtenthaler and Fischbach [37] report results from a meta-analysis of the longitudinal association between job crafting and work engagement. Their results suggest that this is a reciprocal association, where promotion-focused job crafting and work engagement impact each other mutually across time. However, single empirical studies with longitudinal designs display contradicting results. For example, Tims, Bakker, and Derks [38] found that more job crafting predicted higher degrees of both engagement and job satisfaction over a two-month period. An empirical longitudinal study of job crafting that covered both work engagement and job satisfaction was conducted by Hakanen, Peeters, and Schaufeli [39]. They reported that work engagement predicted future crafting behavior four years later, whereas job satisfaction did not. They concluded from their study that the relationship between job crafting and employee well-being may be more complex than commonly assumed because whether and in which way employees craft their jobs seems to depend on their current affective state.

Job crafting has been mentioned in the research literature as part of the job-demand resources model (JD-R), in which job crafting has been defined as proactive changes in employees’ job demands and resources [40]. In the JD-R model, social capital can be identified as a social job resource that promote motivation and work engagement. In this model, work engagement is depicted as increasing the likelihood for job-crafting behavior, which in turn increase personal resources that interplay with job resources. Bakker and Demerouti [40] therefore conclude that “engaged people can create their own ‘gain spiral’ of resources and work engagement through job crafting.’’ There are, in other words, support in the research literature for a predictive effect of work engagement on job crafting, which in turn promotes the forming of job resources rather than the other way around. However, in line with Lazazzara et al. [33], one might also, on the other hand, argue that a supportive social context where employees feel safe and accepted promotes employees’ job-crafting behavior.

In light of these empirical findings, we expect that job crafting will have an immediate short-term impact on both job satisfaction and work engagement. Indications that job crafting might be associated not only with social capital but also with work engagement and job satisfaction create the possibility that job crafting plays a mediating role for these variables. Although the reviewed studies examined job crafting in relation to either social capital, work engagement, or job satisfaction, neither of them aimed to establish the mediational role of job crafting for a prospective predictive effect of social capital on work engagement and job satisfaction.

### 1.4. The Current Study

In the current study, we examined the associations between social capital, work engagement, job satisfaction, and job crafting using two-wave data from health-care professionals working in public health care in Sweden. We used a modeling technique that allows for testing cross-lagged panel designs. This is an important feature of the analyses as to meet the criticism that previous studies that used mediation effects had failed to control for both temporal and concurrent covariation, thereby inflating the risk of obtaining a type I error [41]. In line with current knowledge of how mediating effects should best be tested, we used a modeling technique that also allows for examining mediating effects using bootstrapping rather than the Sobel test [42]. Guided by our general aim and our review of the field, we formulated the following hypotheses: Work-group social capital has a predictive effect on work engagement, job satisfaction, and individual job crafting;Job crafting mediates the potential predictive effect of social capital on work engagement and job satisfaction.

## 2. Methods

### 2.1. Study Design and Participants

A prospective two-wave longitudinal design with correlational data was employed in this study, using the same questionnaires at two points in time (T1 and T2), 6–8 months apart. The target sample consisted of 421 employees from 17 public health-care workplaces in Sweden. The workplaces included dental care clinics and care units in hospitals located in two separate healthcare regions in Sweden. Questionnaires were offered to all employees at workplaces where managers had agreed to participate in a larger study on health-promoting leadership. A total of 250 responded to any of the study variables at both T1 and T2, which means there was a 59% response rate. All 250 participants (*n* = 220 women, *n* = 30 men) in the analytical sample were health-care professionals. The range of job titles included nurses (*n* = 78), assistant nurses (*n* = 45), dental nurses (*n* = 32), work therapists/physiotherapists (*n* = 21), dentists (*n* = 15), dental hygienists (*n* = 11), psychologists/pedagogues (*n* = 9), audiologists (*n* = 8), dieticians (*n* = 1) as well as administrative staff (*n* = 11), clinic coordinators/section leaders (*n* = 2), and a technician (*n* = 1). The remaining 16 participants had not stated their job title.

Of the respondents, 59% worked in professions requiring three years or more of post-secondary education. Within this group, nurses, dentists, dental hygienists, work therapists, and physiotherapists were the most common professions. Together they represented 87% of the participants with three or more years of post-secondary education. Of professions requiring fewer than three years of post-secondary education, assistant nurses was the most common professional group, followed by dental nurses, and administrative staff. These three professions accounted for 97% of the participants with fewer than three years of post-secondary education. In terms of professional experience, 55% of all respondents had worked in their current profession for 15 years or more, 14% had worked in their current profession between 8–14 years, and 31% had worked in their current profession for up to 7 years.

### 2.2. Data Collection Procedure

All procedures and measures that were used in this study were approved by the Regional Ethics Review Board in Stockholm (reference number 2014/5:11) before data collection commenced. Data were collected during 2015 as part of a larger project investigating capacity building for health promotion and sustainable workplaces in health care. After approval from the workplace managers, a questionnaire was distributed individually to their employees. The manager for each separate workplace also chose how the questionnaire was to be distributed (e.g., by email or ordinary mail to the employees’ postal boxes). To ensure that participants remained anonymous during the data collection procedure, age was excluded from the requested background information. Employees also received written information saying that participation was completely voluntary, and they could withdraw their participation at any time without any penalty or consequences. When completed, paper questionnaires were returned by the employers to the research leader using prepaid envelopes, whereas web questionnaires were summarized in a database by a survey company before being sent to the lead researcher. Participants received up to two reminders if they had not replied.

### 2.3. Measures

**Social capital.** The concept of within work-group social capital was captured using five items that referred to its cognitive and structural components. The five items were chosen from a measure of social capital at work that was designed by Kouvonen et al. [19], and they were selected because they applied to social capital within the work group. For the purpose of this study, one item referring to collective trust toward the supervisor was modified to deal with self-reported trust concerning colleagues. The items were, “People keep each other informed about work-related issues in the work unit,” “People feel understood and accepted by each other,” “Members of the work unit build on each other’s ideas in order to achieve the best possible outcome,” “People in the work unit cooperate in order to help develop and apply new ideas”, and “I can trust my co-workers.” Participants responded on a 5-point Likert type scale (1 = do not agree at all to 5 = fully agree), and a mean score across items was calculated. The Cronbach’s alphas for the social-capital scale were 0.88. and 0.86 at T1 and T2, respectively.

**Job crafting.** Three subscales—task crafting, cognitive crafting, and relational crafting—each corresponding with the three conceptual definitions of job crafting established by Wrzesniewski and Dutton [32] were used [43]. The stem question, common for all subscales, was “How frequently do you do the following in your job…?” Participants responded on a 6-point Likert type scale (1 = almost never to 6 = very often), and a mean score across items was calculated. Task crafting was captured using five items that focused on behaviors where the respondents intended to adjust the actual work task to suit their personal preferences. These items were, “introduce new approaches to improve your work,” “choose to take on additional tasks at work,” “change the scope or types of tasks that you complete at work,” “introduce new work tasks that better suit your skills or interests,” and “give preference to work tasks that suit your skills or interests.” Cognitive crafting was captured by five items that focused on the thoughts and mindsets that the respondent attached to personal and the general purpose of the job. These items were, “think about how your job gives your life purpose,” “remind yourself about the significance your work has for the success of the organization,” “remind yourself of the importance of your work for the broader community,” “think about the ways in which your work positively impacts your life,” and “reflect on the role your job has for your overall well-being.” Relational crafting was captured by four items, which focused on behaviors that developed respondents’ social networks at the workplace. These items were, “make an effort to get to know people well at work,” “organize or attend work-related social functions,” “organize special events in the workplace (e.g., celebrating a co-worker’s birthday),” and “make friends with people at work who have similar skills or interests.” For T1/T2, the Cronbach’s alphas were 0.71/0.76, 0.86/0.90, and 0.72/0.75 for the job crafting types focusing on tasks, cognition, and relationships, respectively.

**Work engagement.** The scale of work engagement and burnout (SWEBO) was used to measure work engagement [44]. The SWEBO comprises 10 items covering the three concepts of vigor, dedication, and absorption. The stem question was, “How often during the last two weeks have you at work felt…?,” and the items were “energetic,” “persistent,” “active,” “pride,” “dedicated,” “inspired,” “fully concentrated,” “attentive,” “nimble-witted” and “clear-headed.” Participants responded on a 5-point Likert type scale (1 = not at all to 5 = all the time), and a mean score was calculated. The Cronbach’s alphas were 0.89 for both T1 and T2.

**Job satisfaction.** A subscale of the Copenhagen Psychosocial Questionnaire [COPSOQ; 16] was utilized as the job satisfaction scale for this study. This scale covers six items that capture employees’ overall satisfaction with their work situation, including work environment, management, professional challenges, optimal use of knowledge, and future prospects. The stem question was “How satisfied are you with…?” and the items were, “your prospects in your work,” “the work environment,” “the way your department is managed,” “the ways your knowledge is used,” “the challenges and competence demands in your work,” and “your work in total, all aspects included.” Participants responded on a 4-point Likert scale (0 = very unsatisfied, 33 = unsatisfied, 67 = satisfied, to 100 = very satisfied), and a mean score across items was calculated for overall job satisfaction. The Cronbach’s alphas were 0.88 and 0.86 for T1 and T2, respectively.

### 2.4. Statistical Analyses

Data were analyzed with structural equation modeling (SEM) with manifest variables using Mplus 8.3 [45]. Two separate models were designed in order to test each of the hypotheses. First, the influence of social capital over time on work engagement, job satisfaction, and the three types of job crafting (cognition, work task, and relationship) was examined using a cross-lagged panel design that included both T1 and T2 while controlling for both concurrent and temporal variance. Second, the indirect influence of social capital on work engagement and job satisfaction via the different aspects of job crafting was also examined using a cross-lagged panel model. This model also controlled for temporal variance for all variables and for concurrent variance at T1. There were two-way paths included between all possible combinations of variable pairs. At T2, however, this model controlled for concurrent variance separately for the three variables of social capital, work engagement, and job satisfaction, and separately for the different types of job crafting.

All study variables were approximately normally distributed. Therefore, both models were estimated using the maximum likelihood (ML) estimator. For the second model, we used the bootstrapping technique to perform tests of mediating effects. In addition to the χ^2^ value and the associated degrees of freedom, the model fit was evaluated with the comparative fit index (CFI) [46,47] and the root mean square error of approximation (RSMEA) [48,49]. Although both fit indexes can range from 0 to 1, better model fit is indicated by high CFI values and low RMSEA values. As the probability of type I and type II error for certain fit indexes varies with model complexity and sample size [50], we considered CFI and RMSEA in conjunction with each other in determining certain cutoff values. In line with a suggestion for parsimonious models and relatively small samples [51], we evaluated a combination of a CFI value of 0.97 or higher and an RMSEA value of 0.08 or lower as a good fit in our study.

The analyses controlled for gender effects. In addition, design effects due to potential differences in organizational culture and practice among workplaces (units) were dealt with by first dummy coding the unit variable and then including the resulting binary variables for each of the study variables. This technique has previously been used by Jutengren, Kerr, and Stattin [52] and was chosen rather than multilevel analysis because of the insufficient number of units [53].

Due to internal attrition, 9.2% of the 250 participants did not have complete data on all study variables. Data coverage for any pair of variables ranged from 94.4 to 99.2%. As indicated by the result of Little’s test of MCAR, the requirements for data that are missing completely at random were not violated, χ^2^ (142, *N* = 250) = 162.73, *p* = 0.112. Based on this, to estimate missingness, we could use the full information maximum likelihood (FIML) procedure, which uses all available data from each participant and is considered one of the best practices for handling missing values [54,55].

## 3. Results

An examination of bivariate intercorrelations between study variables revealed a robust pattern of associations, indicating that the SEM analyses would resolve reliably (Table 1). There was no statistically significant (*p*s > 0.05) change in mean value across time for any of the constructs. Parameters in the results are all presented as standardized estimates with two-tailed *p*-values and were tested using an alpha level of 5%.

### 3.1. Main-Effect Model: Direct Effects of Social Capital

The first hypothesis regarding the direct predictive effects of social capital on work engagement, job satisfaction, and job crafting was evaluated using a two-wave cross-lagged panel design. This model controlled for temporal variance across time for all concepts and for concurrent variance among all study variables at both T1 and T2. The estimation of this model yielded a satisfying fit to the data, χ^2^ (41, *N* = 250) = 68.68, CFI = 0.983, RMSEA = 0.052 (90% CI = 0.029–0.073). The results for separate parameter estimates (Figure 1) show that social capital had a positive predictive effect on work engagement (β = 0.12, *p* < 0.05), job satisfaction (β = 0.12, *p* < 0.05), cognitive job crafting (β = 0.10, *p* < 0.05), and relationship job crafting (β = 0.12, *p* < 0.05), but not on task job crafting (β = 0.06, *p* = 0.301). As an additional observation, for concurrent estimates at T2, only three out of nine estimates involving job crafting and any other study variable were statistically significant (Table 2), suggesting that controlling for the concurrent variance for such paths adds little to the validity of the model.

### 3.2. Mediation Model: Indirect Effects of Social Capital via Job Crafting

The second hypothesis related to the indirect predictive effects of social capital on work engagement and job satisfaction mediated by job crafting. This hypothesis was also tested using a two-wave cross-lagged panel design that controlled for temporal variance across time for all included concepts and for concurrent variance among all study variables at T1 (Figure 2). Task-focused job crafting was eliminated from this model because it was not predicted by social capital in the preceding main-effect model. At T2, the concurrent variance was controlled for separately within the three variables of social capital, work engagement, and job satisfaction and within the two variables of cognitive job crafting and relationship job crafting. The model had a satisfying fit to the data, χ^2^ (34, *N* = 250) = 67.650, CFI = 0.978, RMSEA = 0.063 (90% CI = 0.041–0.085). In terms of separate parameter estimates (Figure 2), the results show that social capital had a positive predictive effect on both relationship job crafting (β = 0.12, *p* < 0.05) and cognitive job crafting (β = 0.10, *p* < 0.05). For T2 cognitive job crafting, there were statistically significant positive associations with both T2 work engagement (β = 0.17, *p* < 0.01) and T2 job satisfaction (β = 0.12, *p* < 0.01). In contrast, for T2 relationship job crafting, there were no associations, not with T2 work engagement (β = 0.06, *p* = 0.242) nor with T2 job satisfaction (β = 0.05, *p* = 0.284). Cross-sectional estimates at T1 ranged from 0.17 to 0.59 (*p* < 0.05), and at T2 from 0.24 to 0.48, excluding the path between social capital and work engagement that was not statistically significant (Table 3).

In addition to the separate estimates, we examined both general and specific indirect effects of social capital on work engagement and job satisfaction. First, the general indirect effect from T1 social capital to T2 work engagement was not statistically significant (β = 0.02, *p* = 0.065), and neither were the specific indirect effects through T2 cognitive job crafting (β = 0.02, *p* = 0.120) nor T2 relationship job crafting (β = 0.01, *p* = 0.362). Second, the general indirect effect, this time from T1 social capital to T2 job satisfaction, was not statistically significant (β = 0.02, *p* = 0.106), and neither were any of the specific indirect effects through T2 cognitive job crafting (β = 0.01, *p* = 0.187) and T2 relationship job crafting (β = 0.01, *p* = 0.395).

## 4. Discussion

This study has addressed the predictive influence of social capital and job crafting on work engagement and job satisfaction over time. By testing the hypotheses using two-wave longitudinal data while controlling for concurrent and temporal variance, the results give important support for a causal influence of social capital on both work engagement and job satisfaction. In addition, the results indicate that social capital also promotes some aspects of job crafting. However, there was no evidence for a mediating effect of job crafting.

### 4.1. Effect of Social Capital on Job Satisfaction and Work Engagement

The results revealed a direct positive predictive effect of social capital on both work engagement and job satisfaction. In other words, this result not only reveals cross-sectional associations but also adds to the previous research by providing evidence for a causal effect of social capital on work engagement and job satisfaction. Therefore, by improving interpersonal trust and acceptance among work group members, both work engagement and job satisfaction will increase eventually. In light of the functional differences between the concepts of job satisfaction and work engagement, one might speculate that, although both concepts are empirically affected by social capital, the way in which they are actually influenced by social capital might differ. As job satisfaction, to the larger part, is an outcome of the individual’s retrospective evaluation [18], the conceptual link that explains how it is influenced by social capital is rather straight forward. For most employees, horizontal social capital within a work group is likely to be perceived in retrospect as a favorable aspect of the social work climate and, thus, contributes to a sense of job satisfaction.

For work engagement, with its more prospective way of being evaluated [17], the link from social capital might not be as clearly straightforward. In light of a definition of social capital that emphasizes support, reciprocity, and trust as important factors of horizontal relationships among colleagues [19], one might surmise that social capital transforms into an intentional behavior that influences work engagement through dedication and vigor. For example, one might speculate that an employee who feels his or her relationships with colleagues is mutually supportive, accepting, and trusting would be more likely to engage in work with more dedication and vigor, but this may not necessarily mean they would be more absorbed in the work.

### 4.2. Effect of Social Capital on Job Crafting

The results also revealed a positive predictive effect of within work-group social capital on both cognitive and relational job crafting. The concept of social capital implies that the workplace is characterized by mutual trust and acceptance among individuals [19,21,22]. One might therefore argue that social capital provides a social environment that promotes employees’ personal agency and, in turn, a healthy sense of autonomy in relation to the work situation.

However, there was no evidence of a predictive effect of social capital on task crafting. The differential effects on separate types of job crafting might have to do with the prerequisites that promote or discourage individual employees from taking action to redesign their jobs. For employees to take the opportunity to craft their job situation actively, they first need to perceive their chances of improving their job situation as possible by doing so, whereas lower levels of autonomy and involvement in the job have been found to decrease crafting behavior [32,34]. In this study, participants were employed in the public health-care sector, some of them in professions where the job situation does not leave much room for work task adjustments. Furthermore, within the health-care sector in general, work descriptions are often detailed and firmly enforced [56]. In addition, the work schedules are often tight due to economic cutbacks and streamlined organizations. These circumstances indicate that the prerequisites for task crafting were limited in this study because crafting the actual job tasks requires that the work situation allows for a certain flexibility in terms of work procedures and the execution of working moments.

### 4.3. Mediating Effect of Job Crafting

With respect to the second hypothesis, the results did not reveal any clear evidence for a mediating effect of job crafting between social capital at T1 and work engagement and job satisfaction at T2. Analyzing mediational effects is a way of finding out why, rather than if, two variables (e.g., social capital and work engagement) are associated, thereby disentangling the associations among several variables. We analyzed the mediational effects of job crafting following the advice of Zhao et al. [42], who argue that there is no need to demonstrate that there is a direct association to be mediated first. However, although single estimates involving cognitive job crafting may suggest the contrary, there were neither a general mediating effect of job crafting or any specific effects of separate types of job crafting. In other words, we found no evidence that the link from social capital to work engagement and job satisfaction can be explained by a chain of events where the employee involves in job crafting as a result of an accepting and trustful social climate and that her or his job-crafting behavior, in turn, is the actual reason for being engaged and satisfied with the job.

Health-care employees represented in this study often experience a strictly controlled and straining work situation, such as time pressure while performing tasks within standardized work processes, and it is possible that job crafting in this context loses its favorable results for the employee. Positive aspects of job crafting under these circumstances could be being replaced by a sense of burden of having the responsibility for crafting their own work on top of their already hectic jobs. These circumstances may explain why the results did not reveal any mediating effects of job crafting. Thus the concept of horizontal social capital is limited to its scoop and that interpersonal trust and acceptance relevant for workers’ tendency to craft their job also involves relationships with persons outside the work group, such as managers and supervisors.

### 4.4. Strengths and Limitations

The implementation of this study was designed to optimize internal validity. Therefore, contrary to most previous studies in the field, this study relies on a longitudinal design. This is a necessary requirement to study causal relationships and draw conclusions about one variable’s predictive effect on another over time. In addition, we analyzed data using a panel design. This statistical technique prevents temporal variance from inflating the associations between the study variables. Contrary to a large number of previous studies [41], we also applied this technique when testing for mediational effects. Furthermore, following the advice of Zhao et al. [42], we examined mediational effects using the bootstrapping technique rather than the Sobel test because the latter may increase the risk of type-I error, thereby providing false evidence for mediational effects. Another strength of this study was the use of prospective data. Using data that refer to participants’ current experiences at each separate wave of the data collection reduces the level of recall error that would be present in a study relying on the participants’ retrospective responses.

This study also has some limitations that should be considered when the results are interpreted. First, although the employed design provides evidence for causality that is more than speculative and more valid than cross-sectional studies, including only two-waves in a longitudinal study cannot dispel the risk that confounding variables have influenced the associations. Another limitation has to do with statistical power. Because of the relatively small sample size, we cannot preclude the risk of type-II error. Likewise, using latent, rather than manifest variables would have reduced the risk of type-II error. In other words, due to reservations in terms of statistical power, there might be associations that our analyses did not reveal as statistically significant. Third, as this study did not control for age statistically, and some research has shown that higher age may, under certain circumstances, be associated with more incentives and less restrictive attitudes towards job crafting [57], we cannot be certain whether the three types of job crafting are equally influenced by social capital. The associations involving task crafting in particular may, therefore, display lower variance and, subsequently, weaker associations with social capital compared to cognitive and relational job crafting.

### 4.5. Implications

This study contributes to the existing knowledge about the mechanisms that are related to promoting job satisfaction and work engagement. The JD-R-model [58] is commonly used for identifying working conditions that promote motivation and contribute to positive outcomes for both the individual employee and the organization. Previous research based on the JD-R model has identified different job resources, such as leadership, role clarity, and employee development opportunities, as significant contributors to job engagement and job satisfaction [58,59]. In this context, the results of the current study suggest that social capital is a critical job resource at the work-group level.

It has been argued that a differentiation between hindering demands and challenging demands would be an important addition to the JD-R model [39,60,61]. Hakanen and his colleagues [39] reasoned that employees’ job crafting may have different outcomes depending on the type of demands they are facing. In this context, our results add to the research by highlighting that social capital is a prerequisite for certain types of job crafting but not for others. However, more research in other organizational contexts, besides the public health-care sector, is needed until strong conclusions can be made about the importance of social capital for task crafting.

There are several practical implications for organizations. For example, when planning and organizing work, managers should acknowledge the importance of social capital within work groups. By doing so, they may not only promote a socially resourceful environment for their subordinates but also enjoy the additional advantages that are associated with job satisfaction and work engagement. There are research showing that both social capital and work engagement can be promoted through a relationship-oriented leadership focusing mutual trust, recognition of individual employees and reciprocity between managers and employees [8,25]. Such evidence emphasize the importance of organizations investing in developing a positive social climate. It also implies the importance of setting aside time for reflections and dialogue in how managers and employees can collaborate in crafting work tasks that fit both individual skills and motives and organizational needs.

### 4.6. Suggestions for Future Research

This study suggests that job crafting does occur, and at least some aspects of job crafting increase as a result of preceding higher degrees of within work-group social capital. As many work tasks within the public health-care sector are standardized for quality control and productivity improvement, this may limit the employees’ opportunities to craft their job in general, and their actual job tasks in particular. Such work context might explain why both cognitive and relational job crafting may vary with changing degrees of social capital, whereas task-related job crafting does not. However, knowing that some aspects of job crafting occur as a result of more social capital, it becomes an interest to explore the “hows”, “whys”, and “whens” of job crafting further. For example, more senior employees, who experience more formal autonomy and power, have a greater tendency to perceive job crafting as a challenge within their own expectations of how work should be conducted, whereas more junior employees tend to emphasize their job description and what others expect from them [30]. To expand our knowledge beyond such associations, we need to ask specific qualitative research questions about the manners in which task crafting occurs. For example, do senior employees redesign their job tasks by adding, reducing, or rearranging the order of the separate tasks?; what motives for either job-crafting behavior are senior and junior employees referring to in doing so? From a wider perspective, we also suggest that the following questions should be addressed: (1) what other characteristics than social capital (e.g., position, needs, and desires) have significance for employees to craft their work?; (2) when does job crafting promote well-being?; and (3) are there any differences in job-crafting behaviors between different health-care professions and what are the implications? Further research inspired by these questions has the potential to add to the knowledge of how employee engagement and satisfaction can be promoted in a wider range of workplaces. In addition, our results show that social capital promotes at least some aspects of job crafting. Analogous results have been reported in a meta-analysis of qualitative studies [33]. However, the JD-R-model implies a reversed connection [40]. Taken together, these sources of knowledge open the possibility that the connection between social capital and job crafting is reciprocal. In light of the results of this study, we consider the connections between social capital, job crafting, and work-related well-being as an interactive system to be explored further.

## 5. Conclusions

By providing substantial evidence for a predictive effect on both work engagement and job satisfaction, this study further establishes social capital as an important workplace characteristic. Social capital creates synergy, meaning the sum of what can be accomplished as a group is greater than for each group member in isolation. This study also provides longitudinal evidence that social capital has a predictive effect on cognitive and relational crafting, meaning that social capital also has the potential of promoting employees’ capacity to actively redesign their jobs for a better person–job match.

In light of the results of this study, we believe that the favorable aspects of social capital overshadow the potentially “dark side of social capital” [29]. Justice, mutual collaboration, and trust should be desirable characteristics for any work group, and for most work groups, they far outweigh any potential negatives. Therefore, it would be beneficial for the health-care sector to consider setting up organizations to promote social capital within work groups. Individual workers would benefit in terms of well-being and, as a long-term result, the organization is likely to benefit in terms of efficiency and lower turnover rates.

## Figures and Tables

**Figure 1 ijerph-17-04272-f001:**
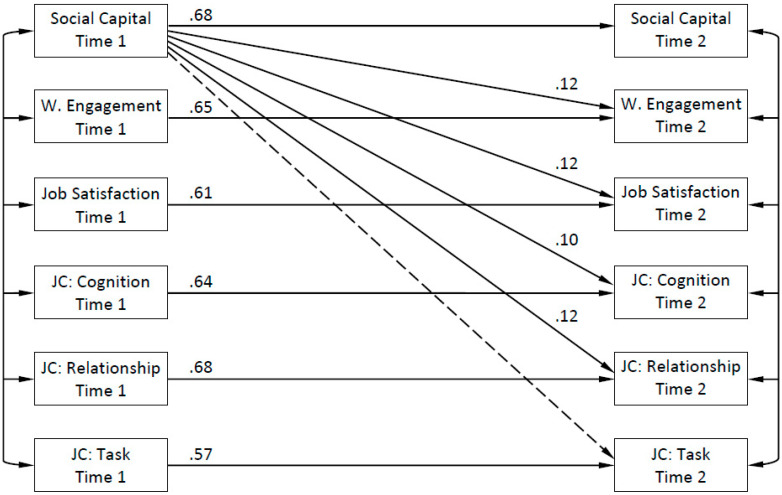
Main-effect model (where W. means work, and JC means job crafting). Dashed line indicates that the relationship was not significant at *p* < 0.05.

**Figure 2 ijerph-17-04272-f002:**
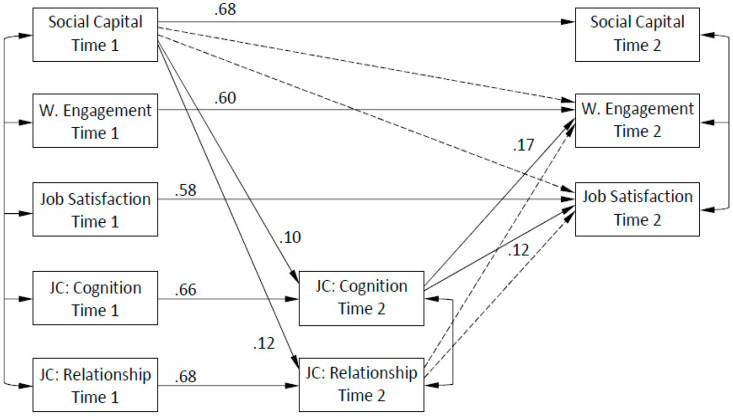
Mediation model (where W. means work, and JC means job crafting). Dashed lines indicate that the relationship was not significant at *p* < 0.05.

**Table 1 ijerph-17-04272-t001:** Intercorrelations, mean values (M), and standard deviations (SD) of study variables (*n* = 250).

Variable	*M*	*SD*	1	2	3	4	5	6	7	8	9	10	11
1. T1 Social Capital	4.00	0.72											
2. T2 Social Capital	3.99	0.69	0.69										
3. T1 Job Crafting: Task	3.41	0.89	0.16	0.07									
4. T2 Job Crafting: Task	3.47	0.82	0.14	0.12	0.59								
5. T1 Job Crafting: Cognitive	3.80	1.17	0.18	0.12	0.44	0.30							
6. T2 Job Crafting: Cognitive	3.82	1.09	0.21	0.14	0.32	0.41	0.67						
7. T1 Job Crafting: Relational	3.80	0.99	0.23	0.08	0.40	0.29	0.39	0.35					
8. T2 Job Crafting: Relational	3.83	0.98	0.27	0.18	0.30	0.36	0.22	0.34	0.72				
9. T1 Work Engagement	3.63	0.59	0.36	0.24	0.34	0.20	0.44	0.38	0.26	0.21			
10. T2 Work Engagement	3.64	0.57	0.35	0.30	0.24	0.17	0.35	0.44	0.24	0.26	0.70		
11. T1 Job Satisfaction	65.20	16.13	0.55	0.48	0.30	0.20	0.30	0.24	0.23	0.25	0.59	0.54	
12. T2 Job Satisfaction	63.76	18.07	0.44	0.60	0.16	0.17	0.22	0.28	0.16	0.24	0.38	0.51	0.71

*Note.* T1 and T2 = Time 1 and Time 2, respectively (6–8 months apart).

**Table 2 ijerph-17-04272-t002:** Standardized estimates and *p*-values for concurrent variance in the main-effect model.

Path	Time 1		Time 2	
	β	*p*	β	*p*
Social Capital vs. Work Engagement	0.36	<0.001	0.12	N.S.
Social Capital vs. Job Satisfaction	0.55	<0.001	0.47	<0.001
Social Capital vs. JC: Cognition	0.18	<0.01	−0.02	N.S.
Social Capital vs. JC: Relationship	0.23	<0.001	0.09	N.S.
Social Capital vs. JC: Task	0.15	<0.05	0.06	N.S.
Work Engagement vs. Job Satisfaction	0.59	<0.001	0.32	<0.001
Work Engagement vs. JC: Cognition	0.44	<0.001	0.26	<0.001
Work Engagement vs. JC: Relationship	0.26	<0.001	0.14	<0.05
Work Engagement vs. JC: Task	0.33	<0.001	0.03	N.S.
Job Satisfaction vs. JC: Cognition	0.30	<0.001	0.18	<0.01
Job Satisfaction vs. JC: Relationship	0.23	<0.001	0.14	<0.05
Job Satisfaction vs. JC: Task	0.29	<0.001	0.09	N.S.
JC: Cognition vs. JC: Relationship	0.40	<0.001	0.23	<0.001
JC: Cognition vs. JC: Task	0.44	<0.001	0.32	<0.001
JC: Relationship vs. JC: Task	0.40	<0.001	0.26	<0.001

**Table 3 ijerph-17-04272-t003:** Standardized estimates and *p*-values for concurrent variance in the mediation model.

Path	Time 1		Time 2	
	β	*p*	β	*p*
Social Capital vs. Work Engagement	0.36	<0.001	0.12	N.S.
Social Capital vs. Job Satisfaction	0.55	<0.001	0.48	<0.001
Social Capital vs. JC: Cognition	0.17	<0.01	-	-
Social Capital vs. JC: Relationship	0.23	<0.001	-	-
Work Engagement vs. Job Satisfaction	0.59	<0.001	0.29	<0.001
Work Engagement vs. JC: Cognition	0.43	<0.001	-	-
Work Engagement vs. JC: Relationship	0.26	<0.001	-	-
Job Satisfaction vs. JC: Cognition	0.30	<0.001	-	-
Job Satisfaction vs. JC: Relationship	0.23	<0.001	-	-
JC: Cognition vs. JC: Relationship	0.40	<0.001	0.24	<0.001

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
