# Peer review of "The Potential Importance of Social Capital and Job Crafting for Work Engagement and Job Satisfaction among Health-Care Employees"

_ijerph, 2020, doi:10.3390/ijerph17124272_

Round 1
Reviewer 1 Report
Unable to review due to illness
Reviewer 2 Report
This study examined the relationship between the associations between social capital, work engagement, job satisfaction, and job crafting using longitudinal data in Sweden. The results suggested that social capital was the predictive factor for job satisfaction, work engagement, and cognitive and relational job crafting. As I understand it, this study has some implications for health care management, but from an epidemiological perspective, there were significant vulnerabilities, particularly with regard to methods, as follows.
- Most importantly, there is no information about age of the study subject at all. In addition, information on the distribution of professions and job titles is also important as basic characteristics.
- It is unclear how the 17 facilities were selected and how the target employees were selected. For these reasons, readers could not capture the representativeness of the target population.
- The analyses should also take into account the confounding of age, gender and other important factors. For example, there are a number of studies showing gender differences in SC. Also, job crafting and work engagement may be affected by period of employment or job title. Alternatively, the authors should state a reasonable reason why those analyses were not conducted.
- If dealing with work-group social capital, it is better to conduct an analysis that takes into account multi-level structures (e.g., 17 facilities or work units).
- All questions included in each indicator should be stated.
- The authors should be more careful with the formatting of the manuscript (e.g., uniform font size, line numbering).
Reviewer 3 Report
Review of “The Potential Importance of Social Capital and Job Crafting for Work Engagement and Job Satisfaction among Health-care Employees”
submitted in Int. J. Environ. Res. Public Health (IJERPH-765077)
Date of Review: 2020/04/30
The manuscript entitled “The Potential Importance of Social Capital and Job Crafting for Work Engagement and Job Satisfaction among Health-care Employees” focuses the influence of social capital on occupational well-being (i.e. work engagement and job satisfaction) and job crafting in a sample of Swedish health-care employees. Furthermore, the authors examined the mediating role of job crafting within the relationship between social capital and occupational well-being. The authors used longitudinal data to test cross-lagged effects between the study variables. Since they found cross-lagged effects between social capital and occupational well-being as well as – partly – job crafting, the authors propose the promotion of social capital within organizations. There was, however, no clear evidence for the mediating role of job crafting within the relationship between social capital and occupational well-being.
The submitted manuscript deals with the interesting topic of social capital as a job resource and its positive impact on occupational well-being. One major advantage is the longitudinal design to test for causal effects between the study variables. The study is well-conducted, the manuscript is well-written and structured and it is of interest to the readers of IJERPH. However, even I see merit in the manuscript, some theoretical and methodological issues need to be addressed before publishing.
My concerns are described as follows:
- I invite the authors to reconsider their theoretical assumptions and research question(s). It could be of advantage to focus the manuscript on the relationship between social capital (introduced as a job resource) and occupational well-being. To examine the mediating role of job crafting properly at least three waves would be needed. However, in case the authors want to keep job crafting in their analyses, I strongly recommend to reconsider their theory. In the current version of the manuscript the authors rely on job crafting theory which posits job crafting between social capital and occupational well-being. The job demands-resources (JD-R) framework (see e.g. Bakker & Demerouti, 2017), however, proposes reverse assumptions. In line with the JD-R framework, occupational well-being (i.e. work engagement) fosters job crafting behaviors which, in turn, acquires new job resources (e.g. social capital). This reversed mechanism should at least be discussed by the authors within their theoretical background.
- Since the authors have well-gathered data with valid and reliable instruments they should make the most of these data. I strongly invite the authors to formulate structural equation models (SEM) with latent variables to answer their research questions. The advantage of SEM with latent variables are measurement error-corrected structural models and the estimation of latent (unobserved) variables via observed variables. In the current version of the manuscript it remains unclear why the authors only used structural regression models with manifest variables.
- Even though the authors formulated research questions, the manuscript would firmly benefit from precisely formulated hypotheses.
- To differentiate work engagement and job satisfaction, the authors should rely on the Circumplex Model of Occupational Well-being (Bakker & Oerlemans, 2011), which introduces both states of occupational well-being very precisely.
- The author should examine the assumptions for maximum likelihood (ML) estimator (e.g. multivariate normal distribution) and report their results. If it is not possible to use ML estimator, they should use MLR estimator.
- Within the results section (»ps > .09«; p. 6) the authors should rather use a more intuitive significance level (e.g. p > .05).
- In the methods section (p. 5), the authors claim that they calculated sum mean scores for work engagement. In table 1 they report, however, mean scores for work engagement. They should be consistent about whether they used sum mean scores or mean scores.
- I was rather surprised that the authors claim dedication and vigor as social oriented components of work engagement (p. 10). This needs some clarification.
Round 2
Reviewer 2 Report
I still think the lack of age information in this study is a critical weakness, but the manuscript has been well modified for the other points I have pointed out.
Reviewer 3 Report
Review of “The Potential Importance of Social Capital and Job Crafting for Work Engagement and Job Satisfaction among Health-care Employees”
submitted in Int. J. Environ. Res. Public Health (IJERPH-765077)
Date of Review: 2020/05/19
The manuscript entitled “The Potential Importance of Social Capital and Job Crafting for Work Engagement and Job Satisfaction among Health-care Employees” has improved substantially after the first revision. The authors have tackled most of my concerns. There is, however, one crucial methodological aspect that – in my opinion – has to be addressed necessarily:
As I wrote in the first round of reviewing, the authors have well-gathered data. As we can read in the methods section, they used valid and reliable scales for all of their constructs. All of these (sub-) scales consist of more than four items. With these data, it is very easily possible to formulate a structural equation model (SEM) with latent variables. The SEM should consist of at least three items per latent factor to capture each construct properly. The authors’ answer that their data “were not sufficient for studying separate types of job crafting using latent variables” disapprove their own data which is simply not true. I strongly suggest that the authors reconsider their methods to make the most of their data. I am firmly convinced that they are able to address this comment. However, I think this aspect is indispensable before publishing is possible.
Round 3
Reviewer 3 Report
I still think that the manuscript would benefit from structural equation modeling with latent variables, but I accept the authors' response on my comment.